# Improving the Segmentation Accuracy of Ovarian-Tumor Ultrasound Images Using Image Inpainting

**DOI:** 10.3390/bioengineering10020184

**Published:** 2023-02-01

**Authors:** Lijiang Chen, Changkun Qiao, Meijing Wu, Linghan Cai, Cong Yin, Mukun Yang, Xiubo Sang, Wenpei Bai

**Affiliations:** 1School of Electronic and Information Engineering, Beihang University, No. 37 Xueyuan Road, Haidian District, Beijing 100191, China; 2Department of Obstetrics and Gynecology, Beijing Shijitan Hospital, Capital Medical University, Beijing 100038, China

**Keywords:** ovarian tumor, 2D ultrasound image, image inpainting, lesion segmentation, attention mechanism, GAN, deep learning, medical image analysis

## Abstract

Diagnostic results can be radically influenced by the quality of 2D ovarian-tumor ultrasound images. However, clinically processed 2D ovarian-tumor ultrasound images contain many artificially recognized symbols, such as fingers, crosses, dashed lines, and letters which assist artificial intelligence (AI) in image recognition. These symbols are widely distributed within the lesion’s boundary, which can also affect the useful feature-extraction-utilizing networks and thus decrease the accuracy of lesion classification and segmentation. Image inpainting techniques are used for noise and object elimination from images. To solve this problem, we observed the MMOTU dataset and built a 2D ovarian-tumor ultrasound image inpainting dataset by finely annotating the various symbols in the images. A novel framework called mask-guided generative adversarial network (MGGAN) is presented in this paper for 2D ovarian-tumor ultrasound images to remove various symbols from the images. The MGGAN performs to a high standard in corrupted regions by using an attention mechanism in the generator to pay more attention to valid information and ignore symbol information, making lesion boundaries more realistic. Moreover, fast Fourier convolutions (FFCs) and residual networks are used to increase the global field of perception; thus, our model can be applied to high-resolution ultrasound images. The greatest benefit of this algorithm is that it achieves pixel-level inpainting of distorted regions without clean images. Compared with other models, our model achieveed better results with only one stage in terms of objective and subjective evaluations. Our model obtained the best results for 256 × 256 and 512 × 512 resolutions. At a resolution of 256 × 256, our model achieved **0.9246** for SSIM, 22.66 for FID, and 0.07806 for LPIPS. At a resolution of 512 × 512, our model achieved 0.9208 for SSIM, 25.52 for FID, and 0.08300 for LPIPS. Our method can considerably improve the accuracy of computerized ovarian tumor diagnosis. The segmentation accuracy was improved from 71.51% to 76.06% for the Unet model and from 61.13% to 66.65% for the PSPnet model in clean images.

## 1. Introduction

Medical ultrasonography has turned out to be the preferred imaging technique for many illnesses due to the fact of its simplicity, speed, and safety [1,2,3,4,5]. Two-dimensional gray-scale ultrasound and coloration Doppler ultrasound has been broadly used in the diagnostic tasks of ovarian tumors. Doctors can first perceive the benign and malignant nature of tumors. With the non-stop development and improvement of deep learning [6,7], AI, as a riding pressure for intelligent healthcare, has acquired a massive range of achievements in tasks such as clinical image classification and segmentation [8,9,10,11]. The accuracy of the model additionally relies upon the quality of the dataset [12,13]. There is exceedingly little research on the current use of AI for lesion recognition and segmentation of ovarian tumor diseases. In addition, the effectiveness of AI in processing ovarian-tumor images depends on a large-scale AI dataset. Zhao et al. [14] proposed an ovarian-tumor ultrasound image dataset for lesion classification and segmentation. The dataset consists of a complete of 1469 2D ovarian ultrasound images which are divided into eight categories according to tumor types. The giant majority of the images in the dataset contain annotated symbols, which are overwhelmingly allotted to inside the lesion.

Nevertheless, hidden but crucial trouble has been recognized in practice: most 2D ovarian-tumor ultrasound images incorporate extra symbols. Actually, in clinical operations where ovarian ultrasound images are acquired, the physician will mark the location, size, and border of the tumor in the ovarian ultrasound image, and observe where the lesion is positioned (left or right ovary). Due to equipment factors and the clinical practice environments, the artificially marked component of these aids to image recognition (symbols such as fingers, crosses, dashes, and letters) cannot be separated from the original image. This phenomenon is also widespread in different medical fields [15,16,17,18]. The ideal situation would be to train and test deep learning models using clean images without any symbols in lesion areas.

We observe that these symbols are centered in ovarian tumor lesions, which negatively affects the training of the model to a positive extent, causing the network to focus more on the symbols in the lesions, which in turn reduces the recognition accuracy of ovarian tumors in the clean images and the segmentation accuracy of the lesions. The different types of images in this paper are shown in Figure 1. The original images with symbols were used as the training set, and two different test sets of clean images and original images with symbols were used as a way to discover the impact of symbols on the segmentation accuracy of the model. Figure 2 and Figure A1 exhibit the effects of our experiments. Fewer training epochs are required to segment more accurate lesion regions in images with symbols, and the segmented regions targeted the yellow line roughly. The clean images, on the other hand, required more epochs and reached lower segmentation accuracy. The results show that the symbols in the images provide additional information to the model enhancing the accuracy of segmentation, which is unrealistic in clinical practice. There is little research on this issue, and it is certainly inappropriate to use the marked ovarian-tumor ultrasound images directly to train the segmentation model. Thus, it is critical for the corrupted areas of the images to be painted, so it is significant for healthcare professionals to use clean images for the artificial intelligence-aided diagnosis of ovarian tumors.

Currently, image inpainting in medical images is in the process of booming and has a lot of potential for development. Existing methods are primarily divided into traditional methods and deep learning-based methods. Traditional methods make use of patch-based or diffusion-based methods, the core of which is to use the redundancy of the image itself to fill in the missing areas with low-level texture features of the image. The following four methods are historically used for inpainting: interpolation [20], non-local means [21], diffusion techniques [22], and texture-dependent synthesis [23]. However, ordinary methods cannot learn the deep semantic features of medical images frequently and can not attain excellent results.

Deep-learning-based methods use convolutional neural networks to extract and learn high-level semantic features in the image to guide the model to fill the missing parts. Inspired by EdgeConnect [24], Wang et al. [25] migrated the method using edge information to medical images. This paper details the study of these methods and use of an attention mechanism, a pyramid-structured generator, to enforce the inpainting of thyroid ultrasound images, which automatically detects and reconstructs the cross symbols in ultrasound images. However, this method has some limitations: the cross symbols in the thyroid ultrasound images used in this approach are small and few, and the effect is negative for ultrasound images containing many large symbols; the detected cross symbols are labeled with rectangular boxes, and this approach does not apply to different symbols with irregular shapes; the real background is covered by these symbols, and the restoration areas have no real background, so how to guide the generative adversarial network for training and evaluation, in this case, is a very necessary issue. Wei et al. [26] proposed the MagGAN for face-attribute editing. The MagGAN does this by introducing a novel mask-guided adjustment strategy to encourage the affected regions of each target attribute to be positioned in the generator, using the corresponding attributes of the face (eyes, nose, mouth, etc.). The method is applied to the face-attribute editing task, which requires segmentation of the face’s attributes, which is different from our task. However, the motivation of making the results more realistic by bootstrapping the model is similar.

In addition, various attention mechanisms have been proposed and are broadly used in image processing. These attention mechanisms have been steadily utilized in the image inpainting task. Zeng et al. [27] expanded on this by proposing a pyramidal structure for contextual attention. Yi et al. [28] proposed a contextual residual aggregation of attention for high-resolution images. The spatial attention mechanism was utilized to solve this problem. To acquire results with a clear structure and texture, the Shift-Net model proposed by Yan et al. [29] replaced the fully detailed layer in the upsampling process with a shift-connected layer, through which the features in the background region are shifted to fill in the holes.

Due to the above issues, in this paper, a one-stage generation model based on GANs is proposed, which swaps the regular convolution with fast Fourier convolutions to enhance the image-wide acceptance field of the model and includes a channel attention mechanism to minimize the model’s focus on symbols to fill the holes using effective features. To the best of our knowledge, we are the first to accomplish image inpainting on 2D ovarian-tumor ultrasound images with large and irregular masks, and our approach achieves more convincing results than others.

Our contributions are as follows:We refined 1469 2D ovarian-tumor ultrasound images for irregular symbols and obtained binary masks to establish a 2D ovarian-tumor ultrasound image inpainting dataset.We introduced fast Fourier convolution to enhance the model’s global perceptual field and a channel attention mechanism to enhance the model’s attention to significant features, and the model uses global features and significant channel features to fill the holes.Our model achieved better results both subjectively and objectively compared to existing models while for the first time performing image inpainting without clean images.We use the restoration images for segmentation training, which significantly enhances the accuracy of the classification and segmentation of clean images.

The rest of the paper is organized as follows: Section 2 describes our dataset and model in detail. The associated experiments and results are detailed in Section 3. The conclusions are introduced in Section 4.

## 2. Methodology

### 2.1. Dataset

In recent years, research about ovarian tumors has increased, and researchers have combined ovarian tumor sonograms with deep learning for ovarian tumor classification and lesion segmentation [30,31,32,33]. Most of the 2D ovarian-tumor ultrasound images used in these studies contain symbols, which are broadly allotted to the edges or inner parts of the lesions. We experimentally confirmed the negative effect of these symbols on the classification accuracy and lesion segmentation accuracy of tumors. The MMOTU dataset [14] is a publicly available ovarian ultrasound image dataset. We obtained a 2D ovarian-tumor ultrasound image inpainting dataset based on the MMOTU dataset by refining annotation processing. As shown in Figure 3, the green dashed line in the figure is how the MMOTU dataset is annotated. We labeled the fingers and letters (brown boxes), numbers (blue boxes), and yellow lines (yellow boxes) in the figure on this basis.

With annotation, a corresponding mask for each image is generated, which masks the various symbols in the image. Figure 4 indicates our pipeline. With these annotations, the corresponding mask for each image was generated to build an inpainting dataset containing 1469 2D ultrasound images of ovarian tumors and masks. We performed experiments about image inpainting on our dataset and the effect of image inpainting on lesion segmentation accuracy in the MMOUT dataset.

### 2.2. Implementation Details

In this study, we used a complete, 2D ovarian-tumor ultrasound dataset with 1469 images that we produced, of which 1200 images were used for training and 269 images were used for testing. Arbitrarily shaped masks were used during training and testing. To make certain the equity of the experiments, we generated unique irregular masks for the images used for testing. The inputs in our experiments had two specifications: one specification was 256×256 (h×w), and the other specification was 512×512 (h×w). We trained and tested our model with both image specifications. The Adam optimizer was chosen to optimize the network. We set the initial learning rate to 0.0001, the batch size for training to 16, and the epoch to 1000. In addition to generating masks using our proposed mask generation strategy, we also performed data enhancement operations on the images during training. The framework was PyTorch, and the devices were two NVIDIA GeForce RTX3090Ti.

### 2.3. Proposed Methods

#### 2.3.1. Network Architecture

We propose an image inpainting model based on fast Fourier convolutions (FFCs) with a channel attention mechanism. Figure 5 indicates the details of our model. The images are downsampled by three convolutional layers and then encoded with the aid of nine fast Fourier Convolution Residual Network Blocks. The decoder obtains the inpainting image by predicting the output of the encoder. These inpainting and original images are fed into the discriminator for adversarial training. Traditional fully convolutional models, such as ResNet [34], suffer from slow perceptual-field growth due to a small convolutional kernel size and limited receptive fields. Due to this reason, many layers in the network lack global context, such that the result has a lack of global structural consistency. We replaced the regular convolution with the fast Fourier convolution to solve this problem. In addition, due to the presence of symbols such as yellow dashed lines in the images, we added a channel attention layer to our model to permit the model to focus more on useful features and make the results more realistic. Figure 6 suggests the specified architecture of the Fast Fourier Convolution Block.

#### 2.3.2. Fast Fourier Convolution Block

Regular convolution is mostly used in deep learning models; however, it cannot capture the global features. Fast Fourier convolutions [35] can be an appropriate solution to this problem. The FFCs divide the input channel into local and global paths: the local path uses regular convolution to capture local information; the global path uses the real fast Fourier transform to obtain information with a global receptive field. The fast Fourier change consists of the following five steps:Transforming the input tensor to the frequency domain using the real fast Fourier transform: $R^{H\times W\times C}\to C^{H\times \frac{W}{2}\times C}$.Concatenating the real and imaginary parts in the frequency domain: $C^{H\times \frac{W}{2}\times C}\to R^{H\times \frac{W}{2}\times 2C}$.Obtaining convolution results in the frequency domain through the ReLU layer, BatchNorm layer, and 1 × 1 convolution layer: $R^{Hand2\times 2C}\to R^{H\times \frac{W}{2}\times 2C}$.Separating the result of frequency domain convolution into real and imaginary parts: $R^{H\times \frac{W}{2}\times 2C}\to C^{H\times \frac{W}{2}\times C}$.Recovering its spatial structure using Fourier inverse transform: $C^{H\times \frac{W}{2}\times C}\to R^{H\times W\times C}$.

As shown in Figure 6, we add a squeeze-and-excitation (SE) layer after the spectral transform block, which performs the squeeze, excitation, and reweight operations in turn. The SE layer automatically acquires each feature channel’s weight via learning, then boosts the beneficial features and suppresses the ones that are no longer beneficial according to the weight. By using the SE layer, we make the model focus more on the useful features rather than on the features of these symbols in the image. Finally, the output of the local and global paths are merged.

#### 2.3.3. Generation of Masks during Training

The approach of mask generation during training has been extensively mentioned in previous research, and it is crucial for the inpainting effect of the model. In early studies, the generated masks are rectangular in shape [36], centered on the geometric center of the image. Models trained with these masks have bad results for images with non-centered rectangular masks. Therefore, the method of generating masks at random locations [37] in the image during training was proposed, but this method fails to provide effective and realistic inpainting of images with irregular masks. Subsequently, the strategy of randomly generating irregular masks [38,39,40] at random locations in the image has emerged.

There are many symbols in the image that obscure the clean image. If these areas are repaired, the results cannot be evaluated realistically due to the fact there is no clean image. We need to guide the network to learn to use features of the non-symbolic regions to fill holes. In our task, we propose a new mask generation strategy by generating random irregular masks at random locations outside the symbolic regions in the image. The generation formula for the masks is as follows: (1)$$m=m_{gen}-m_{prior}$$
where $m_{prior}$ is the mask corresponding to the image in the dataset, $m_{gen}$ is the mask generated by the mask generator, and *m* is the final mask.

### 2.4. Loss Function

The loss function in the generation task is essential for the training of the model, and it calculates the distinction between the ground truth and the inpainting image as the loss value. The loss values are back-propagated, and the model is penalized to update the parameters of each layer. In the end, the loss value is reduced, and the result is closer to the ground truth.

Several extraordinary loss functions were used in our task. In our model, the input uses the corrupted image $I_{in}=I_{ori}⊙\left(1-m\right)$, where $I_{ori}$ denotes the original images and *m* denotes the corresponding mask, for which one denotes the missing pixels and zero denotes the existing pixels. The symbol ⊙ denotes the multiplication of the matrix. G denotes the generator, $I_{inp}$ denotes the final inpainting image generated by the model, and the expressions for the inputs and outputs are shown in Equation (Equation 2).
(2)$$I_{inp}=G\left(I_{in}\right)$$

The perceptual loss [41] is derived by calculating the distance between features captured by the pre-trained network Ψ(.) from the generated images and the original images. To enable the network to understand global contextual information, we compute high receptive field perceptual loss [42] using a pre-trained ResNet with global receptive fields. The calculation of $L_{ResNet}$ can be expressed as follows: (3)$$L_{ResNet}\left(I_{ori},I_{inp}\right)=M\left({\left[\Psi_{ResNet}\left(I_{ori}\right)-\Psi_{ResNet}\left(I_{inp}\right)\right]}^{2}\right)$$
where $I_{ori}$ is the original image or the target image of the generator, $I_{inp}$ is the generated image, and M is the operation of calculating the inter-layer mean after calculating the intra-layer mean. $\Psi_{ResNet}\left(.\right)$ is a pre-trained ResNet implemented with dilated convolution.

To make the generated inline images more realistic and natural in detail, we additionally use adversarial loss. The adversarial loss function $L_{adv}$ is calculated as follows: (4)$$L_{adv}\left(I_{inp},I_{ori},I_{in}\right)=\max_{D}E_{x\in X}\left[\log\left(D\left(I_{ori},I_{in}\right)\right)+\log\left(1-D\left(I_{inp},I_{in}\right)\right)\right]$$
where $I_{ori}$ is the target image, $I_{inp}$ is the inapinting image, $I_{in}$ is the corrupted image, and *D* is the adversarial discriminator.

In our total loss, we also use the $L_{1}$ loss and the perceptual loss of the discriminator network $L_{Disc}$ [43]. The formula for the perceptual loss of the discriminator network $L_{Disc}$ is similar to Equation (Equation 2). The $L_{1}$ loss is calculated as follows:(5)$$L_{1}=\frac{1}{N}\sum_{}\left|I_{ori}\left(p\right)-I_{inp}\left(p\right)\right|$$
where $I_{ori}$ denotes the original image, $I_{inp}$ denotes the inpainting image, and *p* represents the pixel at the same location in both images.

Our total losses are calculated as follows:(6)$$L_{total}=\eta_{1}L_{1}+\eta_{2}L_{adv}+\eta_{3}L_{ResNet}+\eta_{4}L_{Disc}$$
where η is the weight of each loss function. Following [36,39,42], we set $\eta_{1}=10$, $\eta_{2}=10$, $\eta_{3}=30$, and $\eta_{4}=100$ in training.

### 2.5. Evaluation Criterion

We used the evaluation metrics of structural
similarity (SSIM) [44], Frechet
inception
distance
score (FID) [45], and learned
perceptual
image
patch
similarity (LPIPS) [46] to measure the performance of our model. In addition, we used the mean
intersection
over
union (mIoU) evaluation metric to measure the accuracy of lesion segmentation results.

The SSIM is calculated between two windows of size H × W. The value of SSIM is between −1 and 1, where 1 means the two images are identical and −1 means the opposite. The closer the value of SSIM is to one, the better the inpainting effect is. The SSIM calculation formula is defined as follows:(7)$$SSIM=\frac{\left(2\mu_{A}\mu_{B}+c_{1}\right)\left(2\sigma_{AB}+c_{2}\right)}{\left(\mu_{A}{}^{2}+\mu_{B}{}^{2}+c_{1}\right)\left(\sigma_{A}{}^{2}+\sigma_{B}{}^{2}+c_{2}\right)}$$
where $\mu_{A}$ and $\sigma_{A}{}^{2}$ are the mean and variance of image *A*, $\mu_{B}$ and $\sigma_{B}{}^{2}$ are the mean and variance of image *B*, $\sigma_{AB}$ is the covariance of the two images, and $c_{1}$ and $c_{2}$ are the constants that maintain stability.

The Frechet inception distance score (FID) is a metric to calculate the distance between the real image and the generated image feature vectors. It uses the 2048-dimensional vector of Inception Net-V3 before full concatenation as the feature of the image to evaluate the similarity of the two sets of images. The value of FID is greater than or equal to zero. A lower score means that the two sets of images are more similar, and the FID score in the best case is 0.0, which means that the two sets of images are identical. The FID calculation formula is described as follows:(8)$$FID={∥\mu_{gt}-\mu_{pred}∥}^{2}+\mathrm{Tr}\left(\Sigma_{gt}+\Sigma_{pred}-2{\left(\Sigma_{gt}\Sigma_{pred}\right)}^{1/2}\right)$$
where $\mu_{gt}$ and $\Sigma_{gt}$ are the mean and covariance matrices of the real image features, $\mu_{pred}$ and $\Sigma_{pred}$ are the mean and covariance matrices of the generated image features, and Tr is the operation to calculate the matrix trace.

LPIPS is used to measure the difference between two images in terms of deep-level features, and LPIPS is more consistent with human perception than traditional methods such as $\ell_{2}$, PSNR, and FSIM. The value of LPIPS is greater than or equal to zero. A lower value of LPIPS indicates that the two images are more similar, and vice versa. The LPIPS calculation formula is defined as follows:(9)$$d\left(I_{gt},I_{pred}\right)=\sum_{l}\frac{1}{H_{l}W_{l}}\sum_{h,w}{∥w_{l}⊙\left({\hat{y}}_{gt-hw}^{l}-{\hat{y}}_{pred-hw}^{l}\right)∥}_{2}^{2}$$
where *l* is the current computed layer; $H_{l}$ and $W_{l}$ are the sizes of the patches; and ${\hat{y}}_{gt-hw}^{l}$ and ${\hat{y}}_{pred-hw}^{l}$$\in R^{H_{l}\times W_{l}\times C_{l}}$ are the outputs of the current layer. The feature stack is extracted from the *L* layers and unit-normalized in the channel dimension. The vector $w_{l}$ is used to deflate the number of active channels and calculate the $\ell_{2}$ distance.

MIoU is a widely used standard metric in semantic segmentation, which calculates the mean of the ratio of intersection and merges sets of all categories. The value is between zero and one. Closer to one means better the segmentation, and closer to zero is the opposite. Its calculation formula is defined as follows:(10)$$MIoU=\frac{1}{k+1}\sum_{i=0}^{k}\frac{TP}{FN+FP+TP}$$
where *k* is the number of categories, TP is the number of true positive pixels, FP is the number of false positives, and FN is the number of false negatives.

## 3. Experiments and Results

### 3.1. Results

#### 3.1.1. Experiments on the Image Inpainting

Figure 7 indicates the effects of our model on the restoration of the symbolic regions in the ovarian ultrasound images. The boundary, texture, and structure have high similarity to those in the original image. The results show that we have flawlessly removed the symbols from the images. Especially in the lesion area, we removed the yellow line while reconstructing the boundary of the lesion and the content filling of the yellow line area very well. This proves the power of our model. Furthermore, we compare our approach with robust baselines that are publicly available on FID, LPIPS, and SSIM metrics. We performed statistical analysis of the inpainting results on 269 images of the test set.

Table 1 suggests the overall performance of each baseline on our dataset, and the values of the three metrics in the table are the means of the test samples. Smaller FID and LPIPS indicate better performance of the model, and a larger SSIM indicates better performance of the model. Table 2 presents the overall performance of each baseline on our dataset, and the values of the three metrics in the table are the variance of the test samples. The size of the input images in the experiment was 256 × 256. In the statistical analysis, we observed that our model outperformed all other comparable models in SSIM, FID, and LPIPS metrics. Our model achieved 0.9246 for SSIM, 22.66 for FID, and 0.07806 for LPIPS. Table 3 suggests that the upper and lower limits of our method surpass those of the other methods for all three metrics at a confidence level of 95%.

Figure 8 indicates the inpainting results for different models (we show more results in Appendix A). A clear distinction can be found in the blue box area. These baseline models use the learned symbol features to generate the symbol regions, resulting in yellow pixels in the restoration regions. In addition, the regions they generate show significant distortions and folds, with unsatisfactory textures and structures. We address this problem by using an attention mechanism for the model to focus on the features of the fee-symbolic region in the image. Fast Fourier convolution allows the first few layers of the network to quickly increase the receptive field, which allows the model to gain a global receptive field faster and increase the connection between global and local features. The model can better use the global and local features to fill the holes, and the results of the restoration will have the same structural and textural features as the original image, including smoother boundaries and more realistic content. By introducing the channel attention mechanism, our model pays more attention to the features of non-symbolic regions rather than the features of symbolic regions and chooses useful features for image inpainting. Thereby, the restored image is closer to the original image in terms of content, and no yellow pixels appear in the restoration region. In the qualitative comparison, our model showed the best authenticity and details in the results, including smooth edges and high similarity to the original images. Our method better reconstructed the edge structure and content of the lesion in the image, which dramatically improved lesion segmentation accuracy.

#### 3.1.2. Ablation Experiments

To verify that our approaches do reduce the capabilities of the model, we designed ablation experiments for the baseline model. The dataset used for the experiments was our inpainting dataset. We used solely FFCs as the baseline in this experiment.
FFCsFast Fourier convolutions have a larger and more effective field of repetition, which can effectively enhance the field of repetition of our model and improve its capability. We performed quantitative experiments on fast Fourier convolution, dilated convolution, and regular convolution. The convolution kernel size was set to 3 × 3, and the expansion rate of the dilated convolution was set to 3. Table 4 shows the scores of different types of convolution. FFC performed the best, and dilated convolution was second only to FFC; however, dilated convolution depends on the resolution of the image and has poor generalization.Mask generationThe types, sizes, and positions of the mask during training impact the generative and generalization capabilities of the model. In our task, we focused on exploring the effect of mask generation location on the model. Regular, irregularly shaped masks will overlap with a variety of symbols in the image, and this part of the region was devoid of realistic background for a realistic inpainting quality assessment. Additionally, we avoided network learning to use the features of these symbols. We compare our mask generation approach with the conventional method, and Table 5 and Table 6 show that our method effectively improves the SSIM, LPIPS, and FID.Attention mechanismFor the network to attenuate the focus on symbolic features in the image and enhance the focus on other features in the real background, we introduced the SE layer. By introducing the channel attention mechanism, our model pays more attention to the features of non-symbolic regions rather than the features of symbolic regions and chooses useful features. By this method, the restored image is more similar to the original image in terms of content and no yellow pixels show up in the restoration region. Table 5 and Table 6 show the effects of the experiments.

#### 3.1.3. Experiments on the Lesion Segmentation

As we noted in the introduction, our aim of inpainting of 2D ovarian-tumor ultrasound images is to enhance the accuracy of currently popular segmentation models such as Unet and PSPnet for the segmentation of ovarian lesions.

Figure 2 and Figure A1 show the negative effect of symbols in the image on the segmentation of the lesion: they make the model focus more on these symbols. These symbols provide additional information such that the accuracy of segmentation of ovarian-tumor images that are completely clean and without symbols is substantially reduced, which is unacceptable in clinical practice. Therefore, we used the inpainting images and the original images as two training sets, and the clean images as the common test set for experiments on lesion segmentation. Figure 9 and Figure A5 confirm that the segmentation accuracy was improved from 71.51% to 76.06% for the Unet [19] model and from 61.13% to 66.65% for the PSPnet [47] model in clean images. Figure 10 indicates the segmentation results of the Unet model using the clean images as a testing set. Our approach appreciably improves the accuracy of lesion segmentation, and the visualization of segmentation is much better for experiments on lesion segmentation with clean images. These experiments confirm our conjecture and our original aim of performing image inpainting.

## 4. Conclusions

In this paper, we proposed a 2D ovarian-tumor ultrasound image inpainting dataset to investigate the effect of prevalent symbols in images on ovarian-lesion segmentation. Based on this image inpainting dataset, we proposed a 2D ovarian-tumor ultrasound image inpainting model based on fast Fourier convolution and a channel attention mechanism. Labeled images are used as a priori information to guide the model to focus on features in the non-symbolic regions of the images, and fast Fourier convolution is used to extend the receptive field of the model to make the texture and structure of the inpainting images more realistic and the boundaries smoother. Our model outperformed existing methods in both qualitative and quantitative comparisons. It received the highest scores in all three metrics, LPIPS, FID, and SSIM, which proves the effectiveness of our model. We used the inpainting images for training and validation with Unet and PSPnet models, which appreciably enhanced the accuracy of lesion segmentation in clean images. This additionally demonstrates the great significance of our study for computer-aided diagnosis of ovarian tumors.

Our study in this paper did not currently use ground truth of lesion segmentation in the dataset, which may further improve the similarity of lesion boundaries in inpainted images. In future work, we will do further exploration on how to apply the edge information of the lesion to the model to make the boundaries more similar to those in the original image and extend our model to other types of medical images—CT, MRI, etc.

## Figures and Tables

**Figure 1 bioengineering-10-00184-f001:**
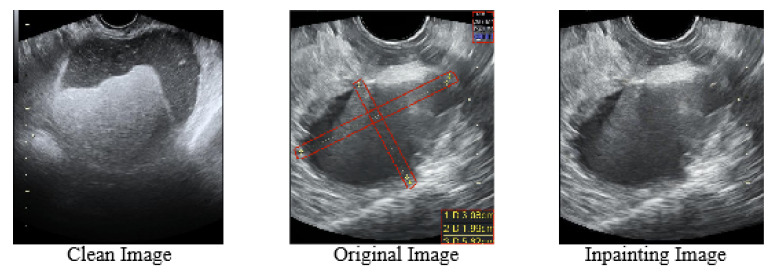
(**Clean Image**) The clean images indicate images that are not clinically labeled. (**Original Image**) The original images indicate clinical images that are labeled. The red-boxed areas show the various marker symbols used by physicians. (**Inpainting Image**) The inpainting images indicate images whose symbols are repaired.

**Figure 2 bioengineering-10-00184-f002:**
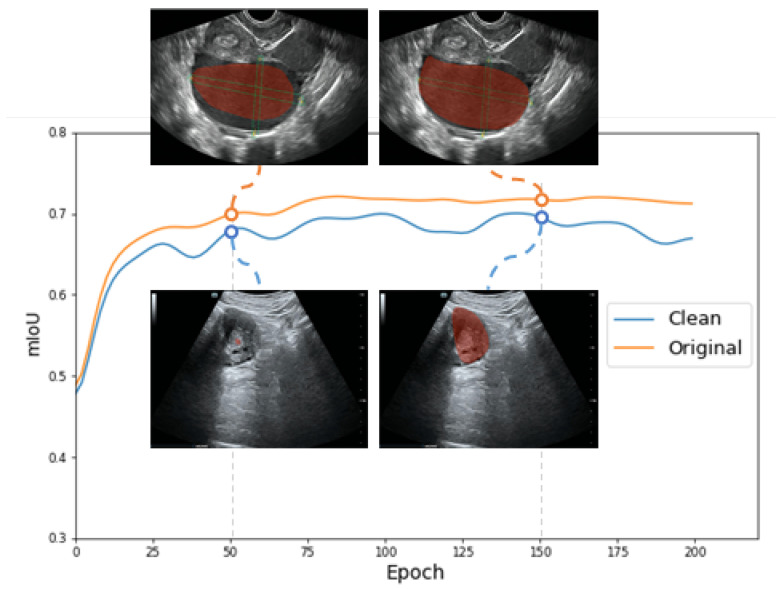
The accuracy graph of lesion segmentation of the Unet [19] model. The blue line represents the accuracy of using the clean images as the testing set. The yellow line represents the accuracy of using the original images with symbols as the testing set. The figure also shows the visualization of the segmentation results for the 50th and 150th epochs.

**Figure 3 bioengineering-10-00184-f003:**
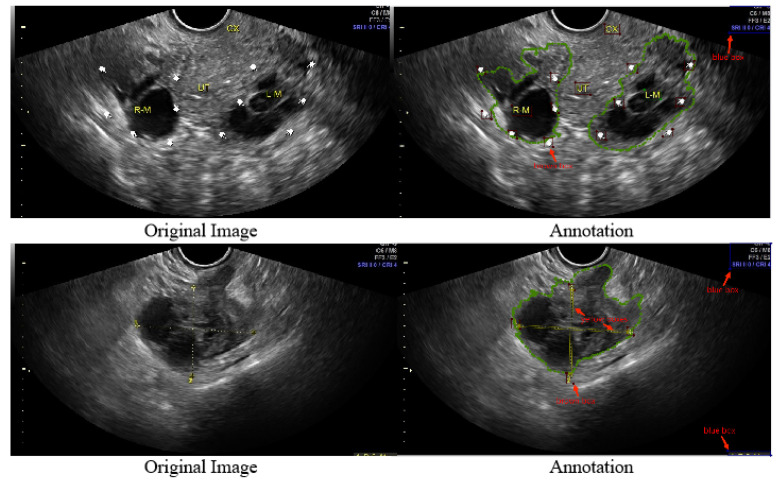
Original 2D ovarian-tumor ultrasound images and images with annotated symbols.

**Figure 4 bioengineering-10-00184-f004:**
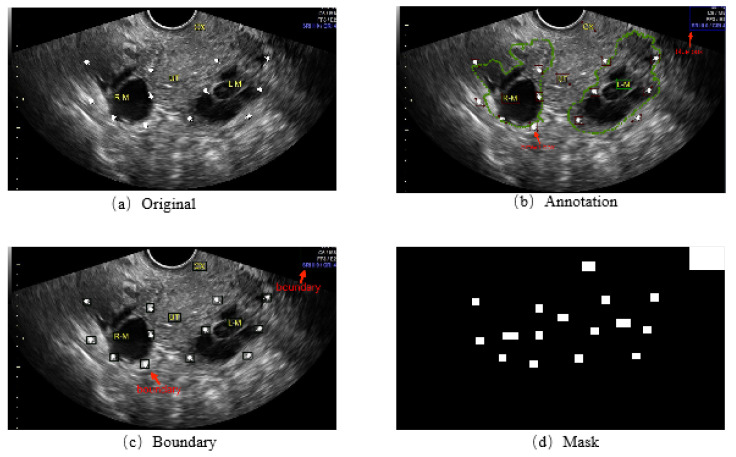
The pipeline of mask generation. (**a**) The original image. (**b**) The annotation. (**c**) The boundary. (**d**) The mask.

**Figure 5 bioengineering-10-00184-f005:**
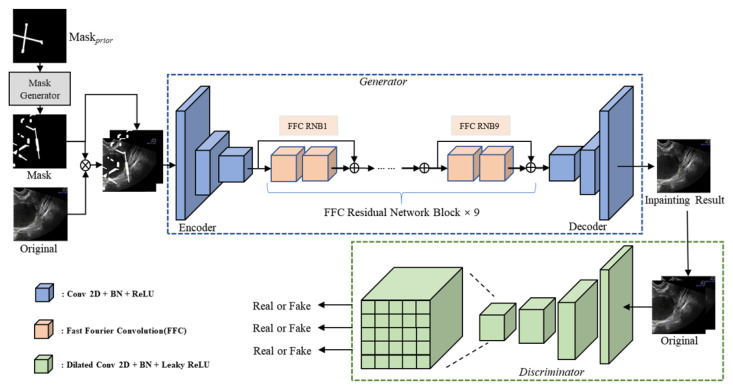
The overall architecture of our MGGAN model. The generator consists of 9 FFC Residual Network Blocks with our mask to a priori guide the generator for image inpainting.

**Figure 6 bioengineering-10-00184-f006:**
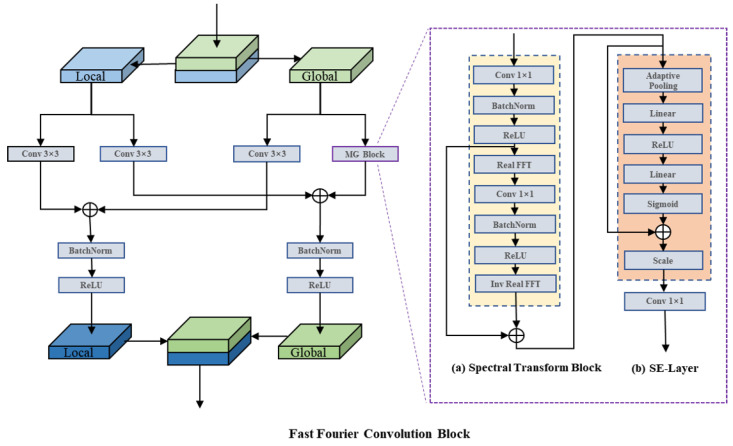
The architecture of the Fast Fourier Convolution Block (FFC Block).

**Figure 7 bioengineering-10-00184-f007:**
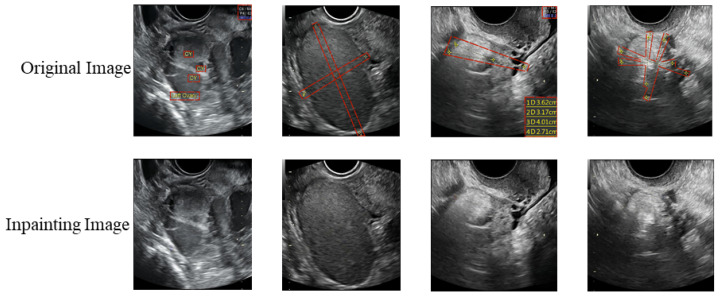
The results of our model for the inpainting of the symbolic regions in the ovarian ultrasound images.

**Figure 8 bioengineering-10-00184-f008:**
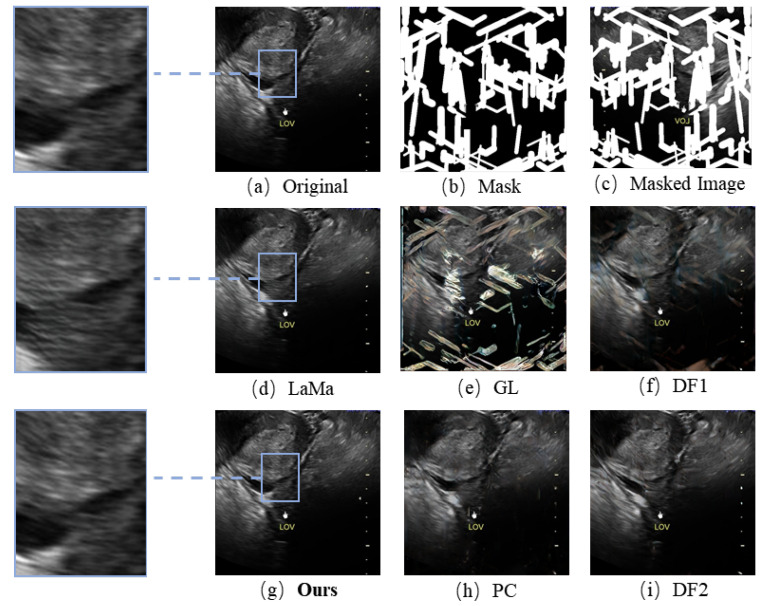
Comparison between the results of our proposed model and other models. (**a**) The original image. (**b**) The mask for the original image. (**c**) The masked image. (**d**) Results from publicly available code using the LaMa method. (**e**) Results from publicly available code using the GL method. (**f**) Results from publicly available code using the DF 1 method. (**g**) Results from publicly available code using our method. (**h**) Results from publicly available code using the PC method. (**i**) Results from publicly available code using the DF 2 method.

**Figure 9 bioengineering-10-00184-f009:**
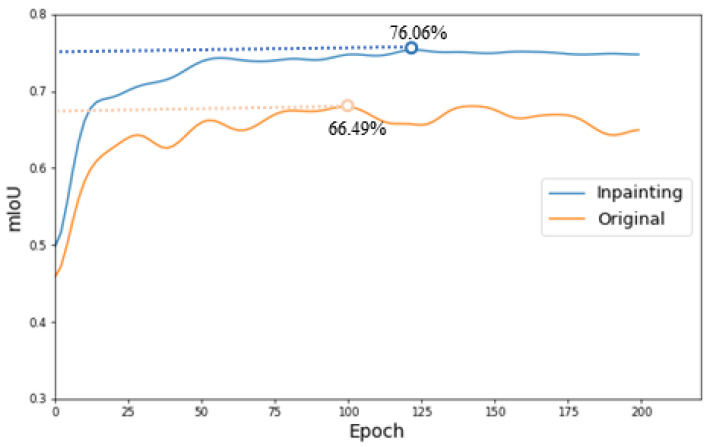
The accuracy graph of lesion segmentation of the Unet [19] model. The blue line represents the accuracy of using the inpainting images as the training set. The yellow line represents the accuracy of using the original images with symbols as the training set.

**Figure 10 bioengineering-10-00184-f010:**
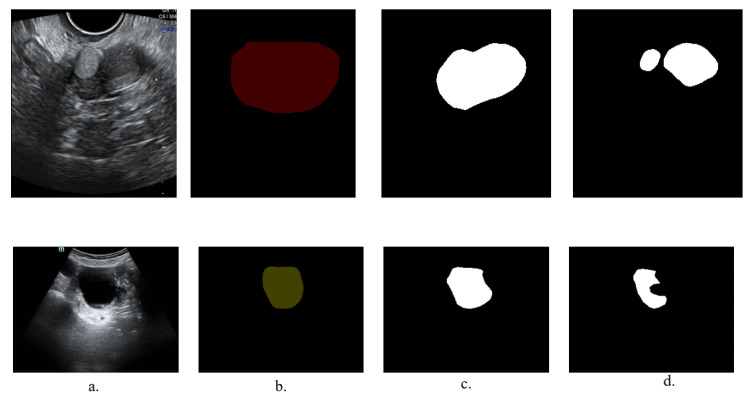
Visualization of the results of lesion segmentation of Unet. (**a**) The clean image. (**b**) The ground truth image. (**c**) Segmentation result of the Unet model using the inpainting images as the training set. (**d**) Segmentation result of the Unet model using the original images as the training set.

**Table 1 bioengineering-10-00184-t001:** Means of the quantitative comparison of the proposed method with already publicly available, robust baselines in FID, LPIPS, and SSIM metrics. The results of each model were derived from its public code.

Model	SSIM	FID	LPIPS
PC [39]	0.6847	79.42	0.13550
GL [38]	0.3026	170.69	0.29589
DF 1 [37]	0.6578	81.74	0.14090
Df 2 [40]	0.8932	54.38	0.10150
LaMa [42]	0.9209	25.54	0.08215
Ours	0.9246	22.66	0.07806

**Table 2 bioengineering-10-00184-t002:** Variances of quantitative comparison of the proposed method with already publicly available, robust baselines in FID, LPIPS, and SSIM metrics. The results of each model were derived from its public code.

Model	SSIM	FID	LPIPS
PC [39]	1.47 × 10^−5^	0.2755	4.5 × 10^−8^
GL [38]	1.81 × 10^−5^	0.4878	9.7 × 10^−8^
DF 1 [37]	1.39 × 10^−5^	0.2801	4.3 × 10^−8^
Df 2 [40]	1.10 × 10^−5^	0.2311	2.1 × 10^−8^
LaMa [42]	9.90 × 10^−6^	0.1777	1.1 × 10^−8^
Ours	9.10 × 10^−6^	0.1373	8.1 × 10^−9^

**Table 3 bioengineering-10-00184-t003:** The lower (left) and upper (right) limits of confidence are 95% of quantitative comparison of the proposed method with an already publicly available, robust baselines in FID, LPIPS, and SSIM metrics. The results of each model were derived from its public code.

Model	SSIM	FID	LPIPS
PC [39]	(0.6771, 0.6923)	(78.17, 80.67)	(0.13510, 0.13590)
GL [38]	(0.2939, 0.3111)	(169.81, 171.57)	(0.29556, 0.29622)
DF 1 [37]	(0.6502, 0.6654)	(80.57, 82.40)	(0.14050, 0.14130)
Df 2 [40]	(0.8860, 0.9004)	(53.46, 55.30)	(0.10122, 0.10178)
LaMa [42]	(0.9145, 0.9273)	(24.72, 26.36)	(0.08195, 0.08235)
Ours	(0.9186,0.9306)	(21.96,23.36)	(0.07788,0.07824)

**Table 4 bioengineering-10-00184-t004:** Effects of different convolutions.

Convs	LPIPS	FID
Regular	0.92230	30.84
Dilated	0.08447	26.77
Fast Fourier	0.08215	25.54

**Table 5 bioengineering-10-00184-t005:** Results of experiments on input with a resolution of 256 × 256.

Model	SSIM	FID	LPIPS
Base (only FFCs)	0.9209	25.54	0.08215
Base + Mask	0.9240	23.08	0.08044
Base + SE-Layer	0.9238	23.02	0.07987
Base + Mask + SE-Laye	0.9246	22.56	0.07806

**Table 6 bioengineering-10-00184-t006:** Results of experiments on input with a resolution of 512 × 512.

Model	SSIM	FID	LPIPS
Base (only FFCs)	0.9170	28.58	0.08939
Base + Mask	0.9189	27.15	0.08842
Base + SE-Layer	0.9102	26.89	0.08769
Base + Mask + SE-Layer	0.9208	25.52	0.08300

## Data Availability

The 2D ovarian-tumor ultrasound image inpainting dataset we created by annotation can be accessed at https://github.com/freedom423/2D-ovarian-tumor-ultrasound-image-inpainting, (accessed on 18 November 2022).

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
