# Peer review of "Improving the Segmentation Accuracy of Ovarian-Tumor Ultrasound Images Using Image Inpainting"

_bioengineering, 2023, doi:10.3390/bioengineering10020184_

Round 1

Reviewer 1 Report

In this paper the authors present a GAN technique based on a channel attention mechanism and fast Fourier convolutions for image impainting via removal of extra annotation symbols from ovarian US images. The proposed technique has the potential to work on other types of images too. The author’s results are novel and relevant. The technique has been thoroughly evaluated on a dataset of about 1500 US images and includes ablations experiments too. After reading the manuscript, here are my comments:

1. Line 59: The two US image pairs in the legend of Figure 2 and A2 correspond to different US images. Although the authors’ conclusion is clear, it would be nicer to show the segmentation outcome on the *same* images. For example, you could keep the ‘original’ pair of images (in the legend) and replace the ‘clean’ image with the ‘clean’ image pair that corresponds to the ‘original’ image pair. This way you’ll be able to show explicitly that during training the symbols indeed affect the IoU when having an original image (i.e. with symbols) as compared to the *same* clean image (without symbols). As it is now, the images shown (above vs below the accuracy graph) are different.

2. Line 170: The ‘brown boxes’ are faint and I don’t see any ‘blue boxes’ and ‘yellow boxes’. I actually see ‘blue letters’, ‘yellow letters’, ‘white letters’ and ‘yellow lines’ (indicating the max dimensions of the ovaries in the 2D plane).

3. Line 269. What’s the actually size of the images in the original dataset? Was it 512x512 and then you performed downsampling to 256x256? What’s the purpose behind the evaluation on 256x256 if US images are typically acquired with higher resolution?

4. The authors should mention the limitations of their study.

I consider that the authors need to address these issues before their paper can be published in the journal

Author Response

Thank you for your hard work and constructive comments! This is a list of responses to reviewers’ comments about the manuscript entitled “Improving Segmentation Accuracy of Ovarian Tumor Ultrasound Images Using Image Inpainting”. 

Reviewer 2 Report

The paper needs some editing, as suggested below.

In a way Figure 2 is the key result, and I wonder if the paper could be re-structured to build up to that, and include different methods of annotation in-paining in the graph.

l89: This sentence is incomprehensible

l93 : more explanation of what in-painting is, and what it is needed for required. In particular it is not clear whether in painting is to replace the annotation or is it painting in the tumour.

l128: what is a look-up hotspot, strange term

l169 (Fig 3) : cannot see brown boxes or blue boxes on image.

Figure 4 : cannot see difference between (b) & (c), could these be emphasised or identified.

L268: How was the training set of images selected, was it  random or another criteria used

L300 What do the values of the metrics SSIM, FID, LPIPS  signify. Is 0.0 good, or 1.0 etc? More explanation required.

L324 What are FFC’s and how do they relate to ablation

L346 What is SE.

Author Response

(The authors gave the same response as above.)

Reviewer 3 Report

Dear Authors,

Your paper is of good quality but is limited to a specific case of ultrasound image treatment based on a novel framework you called mask-guided generative adversarial network (MGGAN).

It seems that methods based on another concept in comparison with methods based on Generative Adversarial Nets (GAN), produce high quality results for image inpainting tasks. 

Could you argue and explain why you did not compare your method MGGAN with methods such as "Image Inpainting Algorithm via Multilevel Attention Progression Mechanism" (cf. Peng Li and Yuantao Chen - Hindawi - Mathematical Problems in Engineering - Volume 2022, Article ID 8508702, 12 pages -  https://doi.org/10.1155/2022/8508702 , for example?

Thank you. 

Author Response

(The authors gave the same response as above.)

Reviewer 4 Report

This manuscript proposes a deep learning approach for inpainting ultrasound images to improve the accuracy of segmentation and classification of ovarian tumors. According to the authors, the proposed network was built by adding the fast Fourier convolution and an attention channel to the Mask-guided generative adversarial network (MagGAN) architecture. The authors used a publicly available database, including 1469 2D ovarian tumor ultrasound images, to train and test their network. The objective of this study was two-fold: 1) to demonstrate that the proposed network had better performance than existing algorithms, and 2) to demonstrate that the proposed inpainting network improved the segmentation outcome. However, the experimental design is problematic: no statistical analysis has been adequately presented in the manuscript. The writing structure needs to be improved. There exist many English usage and inconsistency issues that need to be corrected. The proposed inpainting network should be very interesting to the medical imaging field as marks or annotations are often found in images taken from patients. This reviewer suggests a major revision or a resubmission before considering it for publication. The following list of suggestions may help the authors improve the writing of this manuscript.

1)      Abstract. No segmentation results are shown.

2)      Introduction. Please clearly explain the relationship between the proposed network and MagGAN. An introduction to MagGAN and its application to inpainting should be included.

3)      Introduction. Figure 1: Please choose either “clean image” or “raw image” and make it consistent throughout. Please make sure Figure 2 is necessary. Line 71-84 needs to be shortened.  

4)      Related works. This section can be shortened and included in the “Introduction” section. Section 2.1, Inpainting in Natural Image, is not necessary.

5)      Materials and Methods. 3.1 should replace the title with a more relevant one.

6)      Results. Sections 4.1 and 4.2 should be in the Methods section.

7)      The methods section missed segmentation and its performance evaluation methods.

8)      No statistical analysis is presented, which makes all numbers not meaningful.

9)      What are images for the ground truth? Are any clean images included in MMOTU database? 

Author Response

(The authors gave the same response as above.)

Round 2

Reviewer 3 Report

Thank you very much for your answer and updated version of the paper.